# The Accuracy of Potassium Content on Food Labels in Canada

**DOI:** 10.3390/nu17182935

**Published:** 2025-09-12

**Authors:** Kelly Picard, Dani Renouf, Sarah Draheim, Christopher Picard, Michelle M. Y. Wong

**Affiliations:** 1BC Renal, Island Health Authority, 7-1588 Boundary Cres Nanaimo, Nanaimo, BC V9S 5K8, Canada; 2Dietetics Program, Faculty of Land and Food Systems, University of British Columbia, Vancouver, BC V6T 1Z4, Canada; 3BC Renal, Providence Health, 1081 Burrard St, Vancouver, BC V6Z 1Y6, Canada; drenouf@providencehealth.bc.ca; 4Fraser Health, 13750 96 Ave, Surrey, BC V3V 1Z2, Canada; sdraheim@student.ubc.ca; 5Faculty of Medicine, University of British Columbia, 317-2194 Health Sciences Mall, Vancouver, BC V6T 1Z3, Canada; 6Faculty of Nursing, University of Alberta, 4-171 Edmonton Clinic Health Academy, Edmonton, AB T6G 1C9, Canada; picard@ualberta.ca; 7Division of Nephrology, Department of Medicine, University of British Columbia, 1081 Burrard Street, Vancouver, BC V6Z 1Y6, Canada; mimywong@mail.ubc.ca

**Keywords:** potassium, food labels, hyperkalemia, Canada, label compliance

## Abstract

**Background/Objectives**: In 2017, the Canadian Government updated labeling requirements for prepackaged products to include potassium as a mandatory nutrient. Higher potassium intakes are beneficial in the general population, but for those with hyperkalemia, a lower potassium intake is recommended. **Methods**: The Canadian Food Inspection Agency (CFIA) collects food products and analyzes them to determine their potassium content. The authors requested data collected by the CFIA between January 2005 and November 2023 through an Access to Information request (A-2023-00410). Paired-sample two-sided *t*-tests were used to compare the difference between the labeled and analyzed potassium contents. Cohen’s Kappa was also used to assess agreement between values. **Results**: Data were available for 406 food items, with 376 having a labeled and analyzed potassium value. The number of samples within each product type was not equally spread; 60% of samples (243/405) were considered dairy analogs—comprising either plant-based milk or cheese products. The mean difference between analyzed and labeled potassium content was statistically significant at 15 mg per serving (SD, 68 mg; 95% CI, 8–22 mg; *p* < 0.001). Cohen’s Kappa suggested moderate agreement between labeled and analyzed values (κ = 0.376; 95% CI, 0.305–0.447; *p* < 0.001). A total of 271 (69.7%) products exceeded a ±10% difference, with 90 (23.9%) over-reporting potassium and 181 (48.1%) under-reporting potassium. **Conclusions**: The total number of products that were compliance-tested for potassium in Canada was relatively low and skewed disproportionately toward plant-based dairy analogs. Most products had labeled potassium values that differed from the lab-analyzed values, with a greater tendency to under-report vs. over-report potassium content. This suggests that at least some labels may not be accurate enough to correctly identify high-potassium foods for those who are following a low-potassium diet.

## 1. Introduction

For the general population, higher amounts of potassium in the diet are associated with improved health outcomes, including reductions in stroke and blood pressure [1,2,3,4]. However, for some populations, such as people with chronic kidney disease, heart failure, or diabetes, potassium levels in the blood can become high, a condition known as hyperkalemia. Hyperkalemia can lead to severe outcomes such as cardiac arrhythmias and death [5]. In people with hyperkalemia, limiting dietary potassium intake is often advised [6,7]. One strategy that has been proposed to help people identify suitable foods to meet their health needs is to use nutrition fact tables to identify foods that are high or low in nutrients of interest [8]. Not surprisingly, this strategy is also recommended for people to identify both high- and low-potassium foods. As such, understanding how accurate foods labels are is an essential part of helping people employ this strategy effectively.

The accuracy of sodium content on Canadian food labels has been previously studied using data that is collected from the Canadian Food Inspection Agency (CFIA) [9]. In this study, data collected as part of routine monitoring of nutrition labels by the CFIA was used to determine the accuracy of sodium content on Canadian food labels. In 2016, the Government of Canada amended the Food and Drug Regulations to include potassium per serving size as a mandatory nutrient on the Nutrition Facts Panel [10]. The food industry was initially given a five-year transition period; however, this was extended by another two years due to COVID-19 [11]. Prior to this, including potassium on the Nutrition Facts Panel was voluntary. Given the relatively recent change in the reporting of potassium content, very little is known or has been reported about label compliance monitoring for potassium and how accurate labels are for this nutrient.

Currently, the legislation regarding the accuracy of potassium listed on nutrient labels uses a one-sided limit (minimum value) allowing under-reporting but not over-reporting. As is currently written in the legislation, no specific limit on the degree of under-reporting is provided. This has the potential for harm as people may inadvertently eat items that are high in potassium that are labeled as low in potassium, such as people living with chronic kidney disease. However, given that there are no previous papers written about label accuracy for potassium in Canada, it is currently unknown if this is a real concern or not.

Another factor that may impact potassium nutrient reporting is sodium reduction. Canada has adopted sodium reduction guidelines for use by the Food Industry [12]. In these guidelines, potassium is flagged as an ideal mineral replacement due to its similar taste and chemical characteristics [12]. A previous study reported that products with a reduced-sodium-content claim were more likely to contain a potassium additive, which has the potential to change the potassium content of these foods [13]. Furthermore, many of these potassium additives are highly bioavailable [14], so low-sodium foods that use potassium additives as a sodium replacement and under-report potassium content may increase the risk of hyperkalemia in certain populations (who may choose lower-sodium options to promote health) [13].

Therefore, the primary objective of this study was to describe what efforts have been made to date in Canada with regard to potassium compliance monitoring, specifically looking at differences between labeled and analyzed values for potassium content. As a secondary aim, a sub-analysis of products making a sodium content claim was conducted to determine if label accuracy for potassium differed for products with a sodium content claim.

## 2. Materials and Methods

### 2.1. Study Design

The CFIA samples and analyzes food products to determine their potassium content. Samples are selected by the CFIA based on regular surveillance and monitoring, and to follow up on any consumer or trade complaints [9]. The authors requested data collected by the CFIA between January 2005 and November 2023 through an Access to Information request (A-2023-00410). As this is the first study of its kind, the date range chosen in the information request was arbitrary, as it was unknown when the CFIA started collecting data on the potassium content of foods, or when they started comparing it against labeled potassium. The CFIA data included sample plan ID, laboratory sample number, food category, food group, food subgroup, product class, product subclass, food item, sample description, brand, unit size, label claim—including sodium content claims, potassium, and portion size.

### 2.2. CFIA Compliance Testing

CFIA compliance testing aims to provide a science-based way to verify the accuracy of advertising and nutrient values on labels [15]. According to the CFIA website, the determination of the mineral content of a food is based on 12 individual consumer units taken randomly from a lot. Of these, four units are combined to create three composite sub-samples [15]. Each sub-sample is analyzed, and the mean value of the three sub-samples is used to estimate the nutrient content of the foods [15]. The method of nutrient analysis is selected based on the Official Methods of Analysis, which states that for minerals, including potassium, the correct methods are microwave digestion and inductively coupled plasma mass spectrometry [15]. For naturally occurring nutrients, the CFIA has a stated assumption that a nutrient value must be non-negative and that the coefficient of variation should have an upper bound of 50%. If one of the three lab results is not within 50% of the label value, then this flags a potential concern that the nutrient content is not normally distributed. For added nutrients, a 99% confidence interval is used. The full statistical testing and allowed rounding is provided in the CFIA Nutrition labelling compliance test document [15].

The CFIA compliance criteria for minerals, such as potassium, is that the mean of the three sub-samples must be at least 80% of the declared value if the mineral is naturally occurring, or not less than the declared value if the mineral is added to the food product. The CFIA does not set a specific limit on the amount that potassium may exceed declared values, but instead states that the declared amount overage must be consistent with good manufacturing practice provided that an overage does not present a risk to health and is not misleading. The question-and-answer section of the document further expands on the decision to avoid adopting a two-sided approach (for example, setting an acceptable range for a nutrient between 80 and 120% of the declared value), citing concerns that this range would be too difficult for many manufacturers.

### 2.3. Product Categorization

Of the received data, 133 products did not have data in the product class or subclass category. Products without a product class or subclass were reviewed individually to assign a product class or subclass. Sample description as well as brand and client name were used to identify the correct item and assign the product class and subclass. Items that could not be identified (*n* = 1) were excluded from the product class analysis. All products were assigned a Nova classification [16] for level of food processing based on the product description. Nova is a system of classifying foods by level of food processing, with the levels including unprocessed or minimally processed, such as a whole fruit or vegetable or a cut of meat; processed culinary ingredients, such as oils and sugars; processed foods, such as cheese or bread without additives; and ultra-processed foods, which are likely to contain additives and differ significantly from their unprocessed ingredient counterparts [17].

Sodium content claims were grouped into label claim categories, as defined by the Canadian legislation around content claims [18], including: no added sodium or salt, sodium-free, low in sodium/salt, and reduced sodium/salt.

### 2.4. Statistical Analysis

Paired-sample two-sided *t*-tests were used to compare the difference between the labeled and analyzed potassium contents for the entire sample and to explore differences by over- and under-reporters and by sodium content claim. A *p*-value < 0.05 was considered statistically significant. As it was expected that there would be some level of variability in the potassium content of food items (related to natural variability in the raw ingredients and the level of specificity and precision within tests that determine potassium content), labeled and analyzed potassium values were categorized into five levels: not containing any potassium for products reporting 0 mg of potassium per serving, low potassium (between >0 and 100 mg per serving), moderate potassium (>100–200 mg per serving), high potassium (>200–350 mg per serving), and very high potassium (values above 350 mg per serving) [19]. The agreement between these categorized values was tested with a Cohen’s Kappa test. Additionally, to understand if the December 2016 change in potassium reporting had any impact, a subgroup analysis was performed of samples collected prior to and after 2017.

In accordance with the compliance limit rounding rule standards [15], absolute differences between labeled and analyzed potassium were considered based on the amount of potassium in the product. Products with less than 50 mg of potassium per serving were considered over/under-reporters if the difference was greater than 5 mg. Products containing between 51 and 250 mg of potassium per serving were considered over/under-reporters if the difference was greater than 12.5 mg, and products containing more than 250 mg of potassium per serving were held to a 25 mg difference threshold. Based on a previous similar study analyzing sodium content of Canadian products [9], samples were considered accurate if the percentage difference was less than 20% and were considered inaccurate if the percentage difference was 20% or higher.

A sub-analysis of potassium content and of the accuracy of analyzed potassium content was also conducted for products making a sodium-related label claim. A Chi-Square analysis was performed to test whether there were more inaccurate reports by items in the sodium content claim category with Bonferroni correction applied. An independent-samples Kruskal–Wallis Anova test was used to compare the median difference between labeled and analyzed potassium values, based on the sodium content claim.

## 3. Results

The file received from the CFIA contained 17,665 lines of data (Figure 1). Each food product comprised as many as eight lines of data, including the macro- and micronutrient content, serving size, and label claims. Data from this file was converted to Excel and, for each food item, was amalgamated onto one line for statistical analysis. Not all food items analyzed were checked for potassium content and were therefore not included in this study. This resulted in 406 food items being included in this study.

### 3.1. Number of Products Sampled

Data was available for food items from 2005 to 2023 (Figure 2). For all foods analyzed in 2005 (*n* = 23), potassium was not listed on the label, so these foods were excluded from the analysis of the difference between label values and analyzed values. Additionally, seven other products were missing labeled potassium amounts and were also excluded from the difference analysis. This left 376 products for the analysis of analyzed vs. labeled potassium content.

### 3.2. Types of Products Sampled

Products were grouped into nine broad classes. Additionally, within each class, similar products were grouped into subtypes (Table 1, Appendix A). Products were from a variety of brands (including generic items) and supermarkets across Canada. The number of samples within each product type was not equally spread; 60% of samples (243/405) were considered dairy analogs—comprising either plant-based milk or cheese products.

Regarding the level of food processing, 350 products were considered ultra-processed, 27 were processed, and 2 were processed culinary ingredients (olive oil). There were no unprocessed foods in the data set.

### 3.3. Products Meeting CFIA Compliance Standards

The CFIA’s compliance standard for labeled potassium is that the analyzed (or true) amount must be at least 80% of the labeled value if the potassium is naturally occurring or at least at the labeled value if the potassium is added. This data set does not differentiate between labeled and added potassium, so each product was checked against both of these criteria. In total, 330 (86.6%) products were found to have an analyzed value that was at or above 80% of the labeled value and would be considered compliant if the potassium was naturally occurring. A total of 241 (63.3%) were found to have an analyzed amount higher than the labeled value and would have been compliant if the potassium was naturally occurring or added.

When compliance standards were considered in terms of the absolute potassium content difference between labeled and lab-analyzed products, 73 products were outside of the compliance standard as they had an absolute difference lower than the analyzed value based on portion size (labeled potassium exceeded analyzed potassium by greater than 5 mg for products with potassium content of less than 50 mg per serving (*n* = 21/195), greater than 12.5 mg for products containing between 51 and 250 mg per serving (*n* = 41/133), and greater than 25 mg for products containing more than 250 mg of potassium per serving (*n* = 11/53) (Appendix A provides the threshold compliance limits by product subtype).

### 3.4. Levels of Agreement Between Analyzed and Labeled Potassium Content

The mean labeled potassium content across all food items was 118 mg per serving (SD 192 mg). The mean analyzed potassium content was 134 mg per serving (SD 208 mg). The mean difference (analyzed minus labeled potassium content) was 15 mg per serving (SD, 68 mg; 95% CI, 8–22 mg; *p* < 0.001). Cohen’s κ was run to determine if there was agreement between analyzed and labeled potassium content when considered in terms of zero, low, moderate, high, and very high. There was moderate agreement between the values (κ = 0.376 (95% CI, 0.305–0.447), *p* < 0.001).

### 3.5. Absolute Differences Between Labeled and Analyzed Potassium Content

In total, 192 (50.1%) products were considered to be over- or under-reporting when potassium was considered in absolute amounts, 73 were under-reporting potassium, and 119 were over-reported potassium (Figure 3). Over- and under-reporting was found among all product classes, except Fats and Oils, which only had two products. Beverages had the same number of over-reporting and under-reporting products. Soup had more products that over-reported potassium content on their labels, while the rest of the product classes had more products under-reporting the potassium content on the label.

Fifty items (13.6%) had an analyzed potassium level per serving within +/−1 mg of the labeled potassium amount, with four products (two olive oil, one plant-based cheese, and one cereal product) having an analyzed value that exactly matched the labeled value. Seventy-six items were labeled as containing 0 mg of potassium per serving. Of these, two contained no potassium (olive oil) and 57 plant-based cheese products contained less than 5 mg of potassium per 20–40 g serving. Five items reporting 0 mg of potassium per serving contained over 100 mg of potassium per serving, with two sausage products containing 147 mg per serving and 333 mg per serving of potassium, almonds containing 473 mg per serving of potassium, chicken stock containing 373 mg per serving of potassium, and a cashew-based cheese containing 100 mg per serving of potassium.

### 3.6. Percentage Differences Between Labeled and Analyzed Potassium Content

The number of items within each percentage difference category between analyzed and labeled potassium per serving by product class is displayed in Figure 4. In total, 114 (30.3%) products were within +/−10%; 90 products had a percent difference greater than −10% and were over-reporters, with a labeled value higher than the analyzed value; and 181 had a percent difference greater than 10% and were under-reporters, with a labeled value lower than the analyzed value. Given the uneven sample sizes by product type, no further analysis was conducted at the product-type level.

Across all items, 208 items had a greater than 20% difference between the label value and the lab-analyzed value; of these, 58 were over-reporters and 150 were under-reporters. For the inaccurate over-reporters, the mean potassium content for analyzed vs. labeled potassium was 57 mg per serving vs. 100 mg per serving, with a mean difference of 43 mg per serving (SD, 55 mg; 95% CI, 58–29 mg; *p* < 0.001). For the inaccurate under-reporters, the mean potassium content for analyzed vs. labeled potassium was 111 mg per serving vs. 60 mg per serving, with a mean difference of 51 mg per serving (SD, 85 mg; 95% CI, 38–65 mg; *p* < 0.001).

### 3.7. The Impact of Sample Year on Potassium

Prior to 2017, 139 products with labeled potassium values were tested to determine their potassium content. After 2017, 242 products were tested. As can be seen in Figure 1, prior to 2017, samples were taken from a broad range of product categories; however, after 2017, 226 of the 242 (93.4%) checked products were from plant-based dairy alternatives.

The mean analyzed potassium value prior to 2017 was 260 mg per serving, while the mean labeled potassium value was 241 mg per serving, with a mean difference of 19 mg (*p* = 0.007). After 2017, the mean labeled potassium was 48 mg per serving, compared to 62 mg per serving for the analyzed value, a mean difference of 14 mg (*p* < 0.001). Prior to 2017, three products were labeled as containing 0 mg of potassium per serving, while seventy-three products were labeled as containing 0 mg of potassium after 2017.

Cohen’s κ was run to determine if there was agreement between analyzed and labeled potassium content when considered in terms of zero, low, moderate, high, and very high. Prior to 2017, the agreement was substantive (κ = 0.699 (95% CI, 0.653–0.745), *p* < 0.001) compared to products sampled after 2017, where agreement was low (κ = 0.063 (95% CI, 0.018–0.108), *p* < 0.063).

### 3.8. Sodium-Related Label Claims

In total, 97 products made a sodium-related label claim (e.g., no added salt, low sodium, reduced sodium, or sodium-free) (Table 1). Of these, 10 were from 2006, 17 from 2007, 37 from 2008, 20 from 2009, 1 from 2010, 1 from 2011, 6 from 2018, 1 from 2019, and 3 from 2020. The mean potassium content by sodium content claim and number of products with accurate potassium reporting (within a 20% difference between labeled vs. analyzed) are shown in Table 2. Although products with a sodium content claim generally had higher potassium content than products without a claim, the differences between analyzed and labeled potassium content were similar (mean difference of 14 mg per serving (SD of 60) and *p* < 0.001 for no sodium content claim; 14 mg per serving (SD of 77) and *p* < 0.001 for products with a sodium content claim).

When the type of sodium content claim was considered, only the low-sodium-content claim was found to have a statistically significant difference between labeled and analyzed potassium content; however, given the small numbers, these results should be interpreted with caution. A chi-square analysis between sodium content claim type and number of accurate vs. inaccurate reporters was not conducted, as the minimum number of products within each cell failed to be reached. The Kruskal–Wallis test did not find any statistically significant differences by sodium content claim code or the difference between labeled and analyzed potassium values (H (3) = 3.155, *p* = 0.368).

## 4. Discussion

Our study utilized CFIA data to evaluate differences between labeled and analyzed potassium content in food products in Canada. We found that the total number of products that were compliance-tested for potassium in Canada between 2005 and 2023 was lower than the number tested for sodium, and that a comparison between labeled and analyzed potassium content was not performed in 2005. Most products had labeled potassium values that differed from the lab-analyzed values, with a greater tendency to under-report vs. over-report potassium content.

The number of items that underwent potassium content compliance testing by CFIA between 2006 and 2023 was relatively low at 406, compared to the 1010 items reported to have undergone sodium compliance testing between 2006 and 2010 [9]. Of note, there were several years without any potassium testing or very minimal tests performed. Additionally, the distribution of items that underwent potassium compliance testing was also low, as 60% were plant-based dairy alternatives. While it has been documented that intake of plant-based dairy alternatives is increasing, data from the 2015 Canadian Community Health Survey suggested that only 3% of Canadians consumed plant-based beverages [20]. 

According to the CFIA website, nutrition testing can be performed under two circumstances [21]. The first is routine monitoring, and in this case, the sampling would be unbiased [21]. Second, special or pilot surveys can be undertaken to gather specific information to improve future testing [21]. While the CFIA data files in this study are not labeled with the type of sampling that was conducted, given the high number of foods from dairy analogs, it is possible that these represent special surveys and are not an unbiased sample of the Canadian market place. This poses significant limitations on the generalizability of these results to all food products, as the sample sizes in some product types and subtypes was very small. However, it also highlights how little potassium compliance testing has been performed in Canada to date in the greater marketplace, and suggests a need for greater testing, as there were differences found between labeled and analyzed potassium contents across all product types.

When the samples were divided into those completed prior to and after the 2016 label law changes, there were notable differences. The most striking was that most products tested after 2017 were plant-based milks. Prior to 2017, there was much more variety in the types of product samples. Therefore, while the level of agreement between labeled and analyzed values was substantively different between these two timeframes, it remains unknown if this is a reflection of the types of products sampled or how the food industry is adapting to the legislative changes. These results highlight the need for more comprehensive testing of the food environment to understand how changes in labeling requirements are impacting the accuracy of potassium reporting.

The differences in labeled and analyzed potassium content can be considered in two ways—either an absolute difference (mg per serving) or as a percentage difference. Given the anticipated variability in nutrients of foods, for a variety of reasons, including soil nutrients, time of harvest, and changes during processing and storage, some level of variability is anticipated in the nutrient content of foods [15]. As such, analyses of both absolute and percentage differences were completed. In this study, we found that 192 (50.1%) of products had an absolute difference in potassium content of more than 5 mg if the potassium content was under 50 mg per serving, 12.5 mg if the potassium content was between 51 and 250 mg per serving, and 25 mg if the potassium content was above 250 mg per serving. A total of 69.7% had a percent difference greater than 10%. This discrepancy was found because products which contain very little potassium will yield a higher percent difference. Regardless of the type of difference considered, the majority of products had differences between their labeled and analyzed potassium content. Furthermore, these differences were found across product types, which suggests that it is not one specific product type that is less reliable in terms of potassium content labels.

We are aware of only one other study to have looked at the accuracy of potassium content on food labels. A 2006 study from Australia sampled 350 food items and reported that nearly 40% of samples had a difference greater than ±20% when lab-analyzed values were compared to labeled values for potassium [22]. In Canada, another group compared lab-analyzed values of potassium to the Canadian Nutrient File for meat, fish, and poultry products. In this study, it was reported that 40% of products had significant discrepancies in potassium reporting [23]. The findings from our study are in agreement with these other two studies, in that the potassium content information available to consumers and clinicians through either labels or nutrient databases may be different than the true value.

When there was a difference between labeled and analyzed potassium content, more products were found to under-report potassium on the label compared to the lab-analyzed amount. This finding was anticipated given the current labeling laws. In Canada, a product is considered compliant for potassium reporting provided that the label contains 80% or more of the listed value if the potassium is naturally occurring, or higher than the listed value if the potassium is added [15]. Using either criteria, it is important to note that the majority of products would be considered compliant in this sample.

The current legislation is written with the general population in mind. In the general population, higher potassium intake has shown benefits for both diabetes and endothelial dysfunction [24]. Additionally, Hypertension Canada recommends higher potassium intake to help reduce blood pressure [25]. Evidence suggests that fewer than 40% of adult Canadians meet the adequate intake threshold for potassium [26], and so the risk of under labeling potassium for the majority of Canadians is very low. However, some populations are at risk of hyperkalemia, including those living with kidney disease, diabetes mellitus, or heart disease, and those taking renin–angiotensin–aldosterone system inhibitors (RAASi) [27,28,29]. Among high-risk patients (chronic kidney disease and/or heart failure) in the US, the estimated annual prevalence of hyperkalemia was 6.35% [30], and rates as high as 42% to 73% have been reported for advanced kidney disease [5,31,32]. In these populations, underestimating dietary potassium intake could lead to the overconsumption of potassium and resultant hyperkalemia. In the worst-case scenario, if someone were to inadvertently consume three products that were labeled as containing 0 mg of potassium, but in fact contained between 100 and 473 mg of potassium, as was noted in our results, then this could lead to potential increase in potassium intake by as much ~1200 mg, or 60% of a 2000 mg low-potassium diet. Hyperkalemia is associated with an increased risk of all-cause mortality (an increase of 22% at K 5.5 mmol/L vs. 4.2 mmol/L [33]), arrhythmias (an increase of 59% compared with normokalemia [34]), and major adverse cardiovascular events (an increase of 32% [34]) in patients with chronic kidney disease [32,33,34]. Hyperkalemia has also been associated with reductions in or discontinuation of RAASi, which has also been found to increase the risk of adverse events [35].

Aside from food labels, a key source of information about the nutrient content of foods in Canada is the Canadian Nutrient File, which includes the Nutrient Value of Common Foods, based on the top 1000 consumed foods in Canada [36]. In the Nutrient Value of Common Foods, there are three entries for soy milk, which range in reported potassium content from 308 to 367 mg per cup. However, this current data set includes 13 soy milk products, with an analyzed potassium range of 235–506 mg per cup (average of 376 mg), which suggests that the potassium content will be underestimated for many of these products. Furthermore, the Nutrient Value of Common Foods does not include any plant-based cheese products, which suggests that these products are not part of the top 1000 foods consumed in Canada; however, the CFIA checked 226 of these products, which represents 55.8% of all the potassium content testing performed by the CFIA between 2005 and 2023. While there is limited data from Canada, data from the 2017–2018 National Health and Nutrition Examination Survey (NHANES) in the US suggested that the top sources of potassium were fruits and vegetables and grain-based dishes [37], which again suggests that the current sample of Canadian products is not likely capturing the most important food sources of potassium in Canadian diets. Among patients with end-stage kidney disease on hemodialysis—a population at high risk of hyperkalemia—a US-based study found that the top 10 sources of dietary potassium were beef, chicken, Mexican food, hamburgers, legumes, fresh fruits, real fruit juices, fried potatoes, cheeseburgers, and canned fruits [38]. Few meat products were tested in our data set.

Differences between analyzed and labeled potassium content were explored for products making a sodium content claim vs. products without a sodium content claim. In this subgroup analysis, we did not observe a statistically significant difference. There are several plausible reasons for this finding in our study. First, the total numbers were small, so it is possible that our sample size was too small to detect differences. Second, our data source did not include ingredient lists, so we were unable to assess if products with a sodium content claim were being reformulated using potassium additives, as this has been recommended as a potential strategy by the food industry to lower sodium content [12]. And finally, our results may be impacted by the sample year. All products making a sodium content claim were sampled prior to 2021. The initial sodium reduction document published by Health Canada in 2012 did not explicitly recommend sodium reformulation with potassium [39]. However, in the 2022 updates, potassium chloride is explicitly noted as possible sodium substitute [40]. A future analysis that considers whether or not a product contains a potassium-based sodium substitute is warranted to help understand if the use of these substitutes impacts label accuracy.

Our study has several limitations. The most significant limitations include the small total sample size and lack of representation across product classes and subclasses. That 60% of our samples comprised plant-based dairy analogs significantly limits the generalizability of these results to other product types. Furthermore, these results were limited to Canada and therefore may not be generalizable to other countries. However, this also demonstrates how little potassium compliance testing is being performed in comparison to other minerals such as sodium, and highlights the need for more potassium testing. Given the currently low numbers of products analyzed and the number that were found to misreport potassium content, more widespread compliance testing for potassium content would better help us understand how much potassium is in foods and ensure greater accuracy of listed potassium values.

Another limitation is that the data from the CFIA did not include the ingredient lists of products. Without ingredient lists, we were unable to determine if a product contained a potassium additive and are therefore unable to understand how additives may or may not impact label accuracy. Given that potassium additives, such as potassium chloride, were highlighted in the 2020–2025 sodium reduction guideline [40] as potential sodium substitutes for processed foods, and that potassium chloride is associated with increases in potassium content [19], understanding how these additives may impact label accuracy is important, representing an avenue for future study.

Another limitation is that the data set did not include how labeled values were calculated. In Canada, the industry is given three strategies to generate their nutrient label data, including direct lab analysis or one of two indirect methods. The indirect methods are as follows: a manufacturer can use ingredient databases or use non-specific information from a competitor’s product, or generic information from the literature or what is available in published databases [41]. In a study looking at potassium content listed in the Branded Food Products Database, it was noted that every value in that database used indirect methods [42]. Another consideration when interpreting our results is the serving sizes used by products. There was variability in the serving sizes used to report potassium, even within product subtypes. The 2024 guidelines for food labels in Canada encourage the food industry to use serving sizes that represent the amount of food typically consumed on one eating occasion [43]. As manufacturers update their Nutrition Facts Panels with new serving sizes, the amount of potassium per serving has the potential change and may impact which absolute potassium value threshold each product will be held to. Finally, some products that were analyzed may no longer be available in the Canadian market, as some product samples were 20 years old.

One strength of our study is its novelty. This is the first paper, to our knowledge, to report on compliance testing for potassium; given the changes to the labeling laws, this is likely to continue to be an important study to repeat in future years. Additionally, given that the USA also recently added potassium to nutrition labels, it may be of benefit to repeat this study in other jurisdictions.

## 5. Conclusions

Our study has a very clear implication for practice: Very little potassium compliance testing has been performed in Canada, and that which has been performed is not likely representative of Canadians’ primary food sources of potassium. From this limited data set, however, labeled potassium content was more likely to be under-reported and, in many cases, did not meet the threshold for accuracy that may be required for people using labels to follow a low-potassium diet. Clinicians who work with people with hyperkalemia need to be aware that patients selecting items low in potassium based on the food label may inadvertently consume foods that are high in potassium, which could pose a health risk. On a system level, improved accuracy of potassium content labeling requires action from both food industry and regulatory agencies to ensure that dietary needs of people at risk of hyperkalemia are not overlooked.

## Figures and Tables

**Figure 1 nutrients-17-02935-f001:**
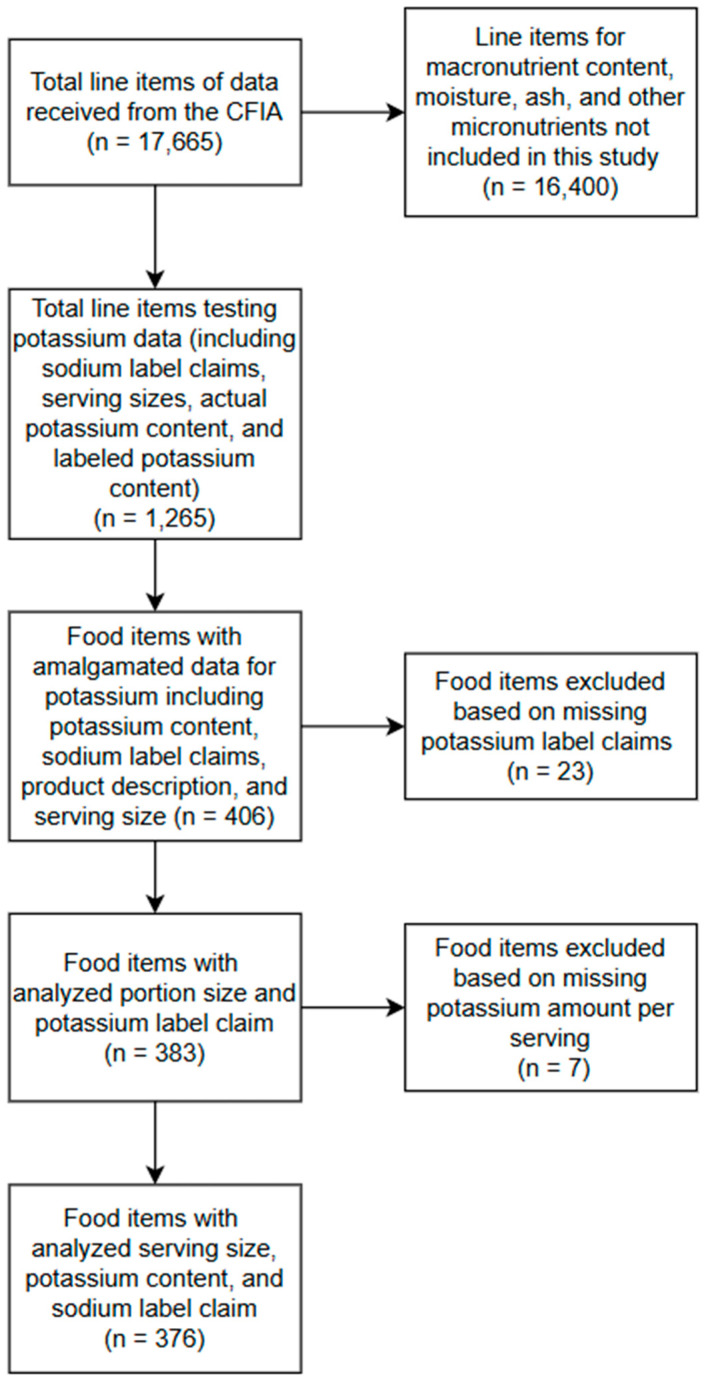
Flowchart representing the data extraction process from CFIA data from January 2005 to November 2023 through Access to Information request (A-2023-00410).

**Figure 2 nutrients-17-02935-f002:**
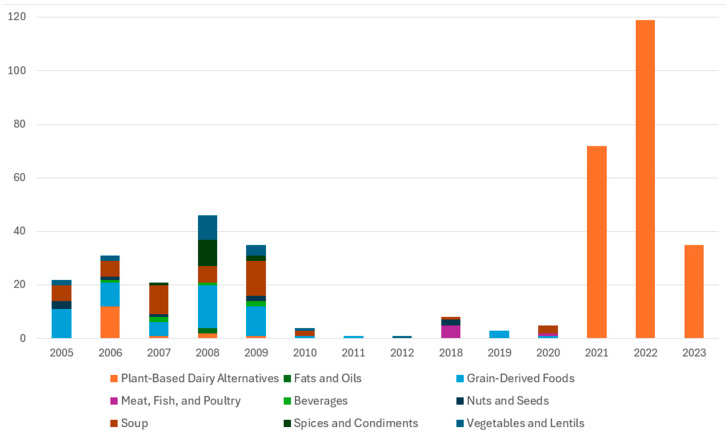
Number of products analyzed to determine potassium content by year and product type.

**Figure 3 nutrients-17-02935-f003:**
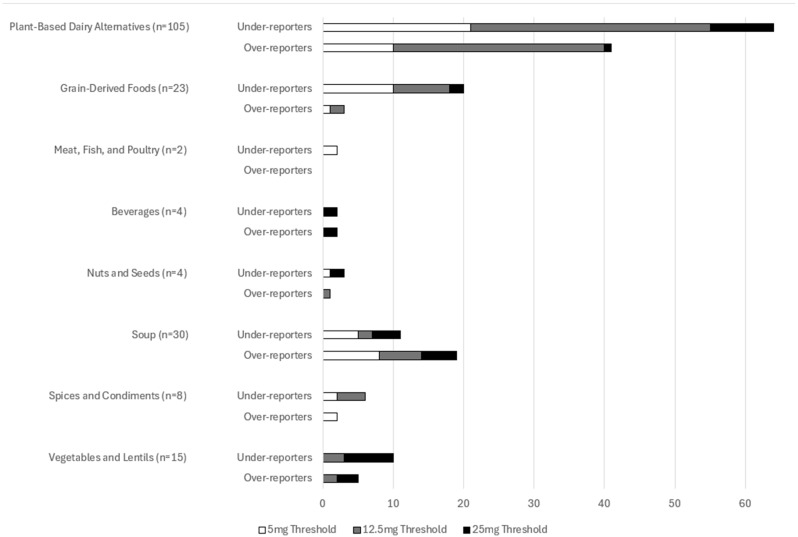
Number of products reporting a difference in potassium greater than 5 mg if it was labeled as containing less than 50 mg potassium per serving, a difference greater than 12.5 mg if it was labeled as containing between 51 and 250 mg of potassium per serving, and greater than 25 mg if it was labeled as containing more than 250 mg of potassium per serving.

**Figure 4 nutrients-17-02935-f004:**
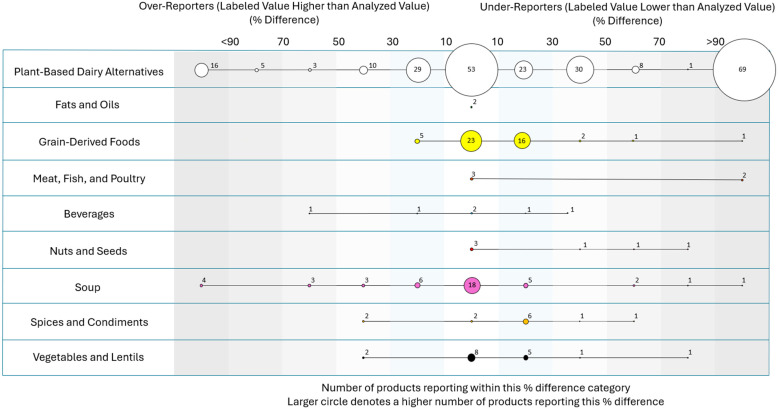
Number of products by product type within each category of percentage difference between labeled and analyzed potassium values.

**Table 1 nutrients-17-02935-t001:** Product classes and subclasses.

Product Class	Product Subclasses
Dairy Analogs (*n* = 243)	Plant-Based Milks (*n* = 16)
	Low sodium (*n* = 2)
	Plant-Based Cheese (*n* = 226) Pudding (*n* = 1)
Fats and Oils (*n* = 2)	Extra Virgin Olive Oil (*n* = 2)
Grain-Derived Foods (*n* = 58)	Cereals (*n* = 15)
	No added salt (*n* = 1) Sodium-Free (*n* = 3) Low sodium (*n* = 4) Reduced sodium (*n* = 1)
	Cookies and Crackers (*n* = 26)
	No added salt (*n* = 1) Low sodium (*n* = 2) Reduced sodium (*n* = 11)
	Pasta and Rice (*n* = 5)
	No added salt (*n* = 3) Low Sodium (*n* = 1)
	Grain-Based Snack Goods (e.g., popcorn) (*n* = 2) Breads and Bread Products (*n* = 10)
	Low sodium claim (*n* = 2) Reduced sodium claim (*n* = 2)
Meat, Fish, and Poultry (*n* = 5)	Sausages (*n* = 2)
	Low sodium (*n* = 1)
	Canned Salmon and Tuna (*n* = 3)
	Low sodium (*n* = 3)
Non-alcoholic Beverages (*n* = 6)	Soda (*n* = 3)
	Low sodium (*n* = 2)
	Juice (*n* = 1) Coconut Water (*n* = 1) Breakfast Drink Mix (*n* = 1)
Nuts and Seeds (*n* = 9)	Peanut Butter (*n* = 4)
	No added salt (*n* = 2)
	Sunflower and Pumpkin Seeds (*n* = 3)
	Low sodium (*n* = 1)
	Almonds and Soy Nuts (*n* = 2)
	Sodium-Free (*n* = 1)
Soups (*n* = 50)	Broth (*n* = 32)
	Low sodium (*n* = 4) No added salt (*n* = 2) Reduced sodium (*n* = 22)
	Canned Soups and Soup Mixes (*n* = 18)
	Low sodium (*n* = 1) Reduced sodium (*n* = 7)
Spices and Condiments (*n* = 13)	Gravy, Sauces, and Condiments (*n* = 9)
	Low sodium (*n* = 1) Reduced sodium (*n* = 5)
	Spice Blends (*n* = 4)
	Sodium-free (*n* = 4)
Vegetable- and Lentil-Based Products (*n* = 19)	Lentils or Beans (*n* = 5)
	Low sodium (*n* = 3) No added salt (*n* = 3)
	Vegetable Juices (*n* = 10)
	Low sodium (*n* = 7) Reduced sodium (*n* = 2)
	Plantain Chips (*n* = 1) Canned Tomatoes (*n* = 1) Seasoned Ground Soy/Simulated Meat Product (*n* = 1)

**Table 2 nutrients-17-02935-t002:** Potassium content by sodium content claim.

Sodium Content Claim	Mean Analyzed Potassium Content, mg per Serving (SD)	Mean Labeled Potassium Content, mg per Serving (SD)	Difference, mg per Serving (SD)	*p*-Value	*N* of Accurate Reporters	*N* of Inaccurate Reports
No Sodium-Related Claim (*n* = 292)	93 (137)	77 (118)	14 (60)	<0.001	120 (41%)	172 (59%)
Sodium Content Claim (*n* = 97)	268 (318)	254 (300)	14 mg (77)	<0.001	55 (62%)	34 (38%)
Low Sodium (*n* = 35)	354 (319 mg)	322 (304 mg)	31 (86)	0.038	25 (71.4%)	10 (28.6%)
No Added Salt (*n* = 8)	442 (513)	428 (540)	13 (40)	0.377	7 (87.5%)	1 (12.5%)
Reduced Sodium (*n* = 46)	173 (245)	171 (210)	2 (73)	0.865	23 (50%)	23 (50%)
Sodium-Free (*n* = 8)	176 (126)	108 (51)	69 (164)	0.275	7 (87.5%)	1 (12.5%)

Table Footnote: SD = standard deviation; mg = milligrams; Accurate Reporters defined as percentage difference being less than 20% between analyzed and labeled potassium content; Inaccurate Reporters defined as percentage difference between labeled and analyzed potassium being 20% or higher. *p*-value determined using paired-sample *t*-test.

## Data Availability

The original contributions presented in this study are included in the article/Appendix A. Further inquiries can be directed to the corresponding author.

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
