# Peer review of "The Accuracy of Potassium Content on Food Labels in Canada"

_nutrients, 2025, doi:10.3390/nu17182935_

Round 1
Reviewer 1 Report
Comments and Suggestions for Authors
Major Comments
This study addresses a critical gap in food labeling compliance by evaluating potassium content accuracy in Canadian prepackaged foods using CFIA surveillance data (2005–2023). The topic is highly relevant given mandatory potassium labeling since 2017 and associated health risks for vulnerable populations (chronic kidney disease). While the manuscript presents novel findings, significant methodological and analytical limitations require resolution before publication.
- The study exhibits significant sampling bias,>50% samples(243/406) are plant-based dairy alternatives despite representing only 3% of Canadian dietary intake (Reference 16). Key categories (meat, poultry, fresh produce) are severely underrepresented . This skews generalizability. Authors must clarify CFIA’s sampling strategy and discuss generalizability limitations.
- Regulatory compliance analysis is inadequate: recalculate the proportion of non-compliant products (labeled K <80% analyzed K, e.g., "sodium-free" sausages containing 333mg K/serving).
- Resolve statistical inconsistencies by standardizing reporting metrics (% difference vs. absolute >5mg difference), redesigning Figure 5 with labeled axes, and justifying potassium categorization thresholds for Cohen’s κ.
- Clinical risks for hyperkalemia patients are noted but not quantified. No linkage between label errors and dietary potassium accumulation(daily intake error from multiple servings). The authors should model worst-case scenarios (daily K intake error if consuming 3 underreported items) and contrast Canada’s one-sided tolerance (allowing underreporting) with the USA’s dual-sided limits.
Minor Revisions
- Specify why 2005 data (pre-mandatory K labeling) was included. Consider sensitivity analysis excluding pre-2017 data
- Define "Nova classification" application since 97% samples were ultra-processed.
- Address why sodium-reduction claims did not significantly impact K accuracy.
- Discuss role of potassium additives using ingredient data if accessible via CFIA.
- Report Bland-Altman limits of agreement (±149mg) in context of serving sizes.
- Table 2: Include p-values for "Any Sodium Claim" vs. subgroups.
- Keywords: Add "Canada," "label compliance."
- Abstract: Highlight "69.7% of products exceeded ±10% error."
- References: Verify publication years.
The study addresses a critical gap in food labeling compliance but requires substantial methodological and contextual refinement to advance the field. Priority revisions are needed before publication
Author Response
Major Comments
This study addresses a critical gap in food labeling compliance by evaluating potassium content accuracy in Canadian prepackaged foods using CFIA surveillance data (2005–2023). The topic is highly relevant given mandatory potassium labeling since 2017 and associated health risks for vulnerable populations (chronic kidney disease). While the manuscript presents novel findings, significant methodological and analytical limitations require resolution before publication.
Response: Thank you for your detailed review. You bring up some excellent points and we value your insight and thoughtful comments. We have responded to each of your points below.
- The study exhibits significant sampling bias,>50% samples (243/406) are plant-based dairy alternatives despite representing only 3% of Canadian dietary intake (Reference 16). Key categories (meat, poultry, fresh produce) are severely underrepresented. This skews generalizability. Authors must clarify CFIA’s sampling strategy and discuss generalizability limitations.
Response: Good point – this wasn’t fully explored in our first version of manuscript. To address this we have added that the CFIA does routine monitoring, but also will follow up on consumer or trade complaints (Lines 87-89). We have also added a statement in the abstract and results section to clearly highlight for the reader the high number of dairy analogues samples (Lin es 25-27, 192-194). Finally, in the discussion we have included a more robust discussion around the CFIA sampling and how this limits the generalizability of our results but also highlights a need for more compliance testing (Lines 325-336).
- Regulatory compliance analysis is inadequate: recalculate the proportion of non-compliant products (labeled K <80% analyzed K, e.g., "sodium-free" sausages containing 333mg K/serving).
Response: Completely agree, thanks for noticing this omission. This has been added to the results section (Lines 200-217). We have also circled back to this result in the discussion (Lines 373-375).
- Resolve statistical inconsistencies by standardizing reporting metrics (% difference vs. absolute >5mg difference)
Response: We have opted to continue to include both a percent and an absolute potassium difference, however, we have redone the analysis on the absolute difference based on CFIA standards and better laid out the difference between these two analysis but reorganizing and renaming these sections in the results (Lines 154-160, Sections 3.5 and 3.6)
- Redesigning Figure 5 with labeled axes
Response: Axes labels have been added to Figure 5.
- Justifying potassium categorization thresholds for Cohen’s κ.
Response: The categorization for Cohen’s κ was reviewed and the decision was made to change the categorization based on a previously published manuscript, which has now been referenced in the methods section. The test was re-run with this binning, which resulted in a slightly lower level of agreement, though the 95% CI were very similar. This was updated in the abstract and results section (Lines 29-30, 147-151)
- Clinical risks for hyperkalemia patients are noted but not quantified. No linkage between label errors and dietary potassium accumulation (daily intake error from multiple servings). The authors should model worst-case scenarios (daily K intake error if consuming 3 underreported items) and contrast Canada’s one-sided tolerance (allowing underreporting) with the USA’s dual-sided limits.
Response: Great point. We have added a section in the discussion to illustrate what consuming 3 products labeled as 0mg of potassium but in fact containing 100-473mg as we found in our results, could mean in terms of a low potassium diet. We have also further detailed the real clinical risks of hyperkalemia (Lines 376-397)
Minor Revisions
- Specify why 2005 data (pre-mandatory K labeling) was included. Consider sensitivity analysis excluding pre-2017 data
Response:
Re: Including 2005 data – this has been added to the study design description (Lines 91-93, 312-313).
Re: 2017 data - Another reviewer also brought up the importance of time, so we have decided to look at our data pre and post 2017, as the new regulations around potassium labeling came out in December of 2016 (which has now been added to the introduction – Lines 55-60). We explained how we did this analysis in the stats section, added it to the results and discussion section (Lines 151-153, 268-284, 337-345).
- Define "Nova classification" application since 97% samples were ultra-processed.
Response: Definition was added (Lines 132-136).
- Address why sodium-reduction claims did not significantly impact K accuracy.
Response: Great point! This has been added to the discussion and we have expanded on our results section to include the sample year of products making a sodium content claim as this likely played a role as well (Lines 287-289, 420-430).
- Discuss role of potassium additives using ingredient data if accessible via CFIA.
Response: We have expanded on this in the discussion section of the manuscript (Lines 444-451).
- Report Bland-Altman limits of agreement (±149mg) in context of serving sizes.
Response: We have decided to remove the Bland-Altman analysis as was suggested by another reviewer.
- Table 2: Include p-values for "Any Sodium Claim" vs. subgroups.
Response: Added in the statistical analysis section and manuscript (Lines 167-169, 301-303).
- Keywords: Add "Canada," "label compliance."
Response: Added (Keywords).
- Abstract: Highlight "69.7% of products exceeded ±10% error."
Response: Added (line 30)
- References: Verify publication years.
Response: All references reviewed, double checked and updated as indicated by this review.
Reviewer 2 Report
Comments and Suggestions for Authors
In nutritional management, potassium intake is considered to be less actively evaluated compared to other nutrients, such as sodium, protein. The authors appear to have focused on this point, but it would be beneficial to provide an overview of the benefits of increasing potassium intake in the general population and the potential disadvantages in specific groups.
The authors state that without accurate ingredient labeling, patients who require potassium restriction may consume more potassium than expected. Patients are likely to refer not only to food labels but also to potassium content in food composition tables. Please explain the relationship between the CFIA data examined by the authors and the Canadian standard food composition tables. This explanation may be included in the discussion.
INTRODUCTION
Please explain the current regulations in Canada regarding the labeling of potassium. Is it mandatory for processed foods, or is it required for foods that meet certain standards? Please specify the units used for labeling (mg, g, per 100 g of edible portion, per serving, etc.).
An explanation of potassium intake levels among Canadians is desired. Furthermore, it is necessary to explain the percentage of the population that requires potassium intake restrictions, such as those with CKD, and the extent of complications thought to be caused by excessive potassium intake among these patients.
Line 83-86: “The CFIA data included: Sample Plan ID, Laboratory Sample Number, Food Category, Food Group, Food Subgroup, Product Class, Product Subclass, Food Item, Sample Description, Brand, Unit Size, Label Claim, Potassium, Portion Size and Label Claim.”
The data should have included sodium content, because the authors compared the foods according to their sodium content. Please add the method for measuring sodium and potassium. Please explain “portion size”. Is it “one bag's worth,” or is it based on a person's nutritional requirements, or is it a customary serving size for one person, because the authors consider cases exceeding ±5 mg of the labelled amount to be over- or under-reporting.
Are potassium values listed on food labels based on measurements taken by manufacturers, or are they calculated using ingredient composition and standard food composition tables?
Line 90: “the nutrient content of three samples of four units”
What’s the “four units”?
Figure 1. Was there any problem with the sodium data?
RESULTS
“3.3 Difference between analyzed and Labeled Potassium Cotent”, and “3.4. Over and Under-Reporting of Potassium Content”
The authors must not only explain the results of the summary in text, but also show the values in tables. In the table, please add the average portion size for the each food category.
Units should be added when reporting the results.
Line 181-182: “147g and 333mg of potassium” should be “147mg/serving and 333mg/serving of potassium”.
Table 2.
Units should not be included in the Table, ie. “93mg” should be “93”.
Abbreviations should be spelled out, and the criteria used for the analysis, and test method used to calculate p-values should be described. Standards for “inaccurate reports” should be described. There are two “N of inaccurate reporters”, which should be under- and over-reports.
Figure 4. Please add the n of foods included in the analysis to the food group names on the vertical axis.
Line 232-233: “was lower than the number tested for other minerals”, please specify the “other minerals”.
DISCUSSION
Please explain the average potassium intake of Canadian people and attributable foods for the total potassium intake. The authors state that it is necessary to verify the difference between label information and actual measurements in order to avoid the risk of people who need to restrict their potassium intake consuming excessive amounts of potassium. Based on the range of foods verified in this study, what percentage of the average potassium intake of Canadians was verified?
Author Response
In nutritional management, potassium intake is considered to be less actively evaluated compared to other nutrients, such as sodium, protein. The authors appear to have focused on this point, but it would be beneficial to provide an overview of the benefits of increasing potassium intake in the general population and the potential disadvantages in specific groups.
Response: Great point, this wasn’t fully discussed in the first version. We have added a paragraph about this to the discussion section (Lines 376-397).
The authors state that without accurate ingredient labeling, patients who require potassium restriction may consume more potassium than expected. Patients are likely to refer not only to food labels but also to potassium content in food composition tables. Please explain the relationship between the CFIA data examined by the authors and the Canadian standard food composition tables. This explanation may be included in the discussion.
Response: We have added a few key examples on how the data that we were provided from the CFIA aligns with the Canadian Nutrient File data, in terms of both potassium content and top consumed foods (Lines 398-417).
INTRODUCTION
Please explain the current regulations in Canada regarding the labeling of potassium. Is it mandatory for processed foods, or is it required for foods that meet certain standards? Please specify the units used for labeling (mg, g, per 100 g of edible portion, per serving, etc.).
Response: Good addition, we have added this to the introduction – a summary of labeling laws around potassium (Lines 55-60).
An explanation of potassium intake levels among Canadians is desired. Furthermore, it is necessary to explain the percentage of the population that requires potassium intake restrictions, such as those with CKD, and the extent of complications thought to be caused by excessive potassium intake among these patients.
Response: We have added a paragraph about intake levels of potassium for Canadians as well as data around the prevalence of hyperkalemia to the discussion section (Lines 379-381 and 384-386).
Line 83-86: “The CFIA data included: Sample Plan ID, Laboratory Sample Number, Food Category, Food Group, Food Subgroup, Product Class, Product Subclass, Food Item, Sample Description, Brand, Unit Size, Label Claim, Potassium, Portion Size and Label Claim.”
The data should have included sodium content, because the authors compared the foods according to their sodium content.
Response: We didn’t look at the sodium content of the foods, only if they made a sodium content claim; to make this more clear, we have updated our Figure 1 and methods section (Line 96).
Please add the method for measuring sodium and potassium. Please explain “portion size”. Is it “one bag's worth,” or is it based on a person's nutritional requirements, or is it a customary serving size for one person, because the authors consider cases exceeding ±5 mg of the labelled amount to be over- or under-reporting.
Response: We have added how the CFIA determines potassium content (Lines 104-107). We have also adjusted how the absolute potassium values were considered based on the potassium per serving size (Lines 154-160) and added in our discussion how serving sizes could be impacting our results (459-465). We also added the serving sizes of products to Supplemental Table 1.
Are potassium values listed on food labels based on measurements taken by manufacturers, or are they calculated using ingredient composition and standard food composition tables?
Response: This highlights an important limitation of our data, we don’t know how manufacturers determined the potassium content. We have added this to the limitation section of our manuscript (Lines 452-459).
Line 90: “the nutrient content of three samples of four units” What’s the “four units”?
Response: This section has been re-written to better explain the compliance testing done by the CFIA (Lines 100-104).
Figure 1. Was there any problem with the sodium data?
Response: We have updated this figure to highlight that we only looked at the sodium content claims as opposed to the sodium content specifically.
RESULTS
“3.3 Difference between analyzed and Labeled Potassium Content”, and “3.4. Over and Under-Reporting of Potassium Content”
Response: The sub-section titles for section 3 have been renamed to be more clear and to accommodate other edits that were made by the reviewers (Lines 218, 226, 249).
The authors must not only explain the results of the summary in text, but also show the values in tables. In the table, please add the average portion size for the each food category.
Response: Median and IQR of serving sizes by product sub types has been added to Supplemental Table 1.
Units should be added when reporting the results. Line 181-182: “147g and 333mg of potassium” should be “147mg/serving and 333mg/serving of potassium”.
Response: The text has been reviewed to ensure that it is clear that the potassium content is being reported per serving.
Table 2.
Units should not be included in the Table, ie. “93mg” should be “93” and the criteria used for the analysis, and test method used to calculate p-values should be described.
Response: mg removed. How p-values were calculated added as a table footnote.
Abbreviations should be spelled out
Response: Abbreviations added to table footnote.
Standards for “inaccurate reports” should be described. There are two “N of inaccurate reporters”, which should be under- and over-reports.
Response: Definition of accurate and inaccurate were included in the table footnote.
Figure 4. Please add the n of foods included in the analysis to the food group names on the vertical axis.
Response: n’s added to each product category.
Line 232-233: “was lower than the number tested for other minerals”, please specify the “other minerals”.
Response: Changed to sodium, as this was our main comparator (Line 312).
DISCUSSION
Please explain the average potassium intake of Canadian people and attributable foods for the total potassium intake.
Response: We have added how many Canadians are not meeting that adequate intake line (Lines 379-380) as well as a discussion on food sources of potassium in the general population (Lines 410-411) and in the CKD population (Lines 414-416).
The authors state that it is necessary to verify the difference between label information and actual measurements in order to avoid the risk of people who need to restrict their potassium intake consuming excessive amounts of potassium. Based on the range of foods verified in this study, what percentage of the average potassium intake of Canadians was verified?
Response: We have added that given the dis-proportionate sampling of plant-based dairy significantly limits generalizability (Lines 434-436).
Reviewer 3 Report
Comments and Suggestions for Authors
Dear Authors,
Thank you for the opportunity to review your manuscript on the accuracy of potassium labeling on food products in Canada. The topic is timely and essential from a public health perspective, particularly for patients requiring a low-potassium diet, such as individuals with chronic kidney disease. The authors conducted a thorough analysis using data from the CFIA, applying appropriate statistical tools, and delivering practical conclusions.
The novelty of the manuscript lies in its use of non-public CFIA data to evaluate the accuracy of potassium labeling—this appears to be the first such study in the Canadian context. An additional strength is the inclusion of a discussion on the potential interactions between sodium-related claims and potassium content. The potassium underestimations identified by the authors could have real clinical implications, thereby justifying the need for further research and potential regulatory review.
At the same time, a few limitations should be acknowledged: the dataset is unbalanced (with over 50% of products being plant-based dairy alternatives), some of the data are over a decade old, and the lack of access to product ingredient lists prevents a deeper analysis of potassium sources (e.g., additives). Therefore, the conclusions regarding the impact of sodium substitutes on potassium content should be interpreted with greater caution.
I kindly ask the authors to consider and address the following suggestions in the revision of the manuscript:
-
Please discuss in more detail the overrepresentation of plant-based dairy alternatives in the sample—can this be corrected in the analysis, or at least acknowledged as a significant limitation?
-
Please clearly differentiate the results derived from older and newer data (e.g., pre-2010) to improve the relevance and credibility of the conclusions.
-
While the references are up to date, you might consider adding studies that assess label accuracy for potassium in other jurisdictions (e.g., the U.S., EU) to strengthen the comparative context.
-
It would be valuable to refer to similar research conducted in other countries (e.g., the U.S., EU) to place the findings in a broader context.
-
Consider expanding the discussion on the practical implications of potassium underestimation—particularly its potential impact on patients treated with RAAS inhibitors.The reviewer.
Author Response
Thank you for the opportunity to review your manuscript on the accuracy of potassium labeling on food products in Canada. The topic is timely and essential from a public health perspective, particularly for patients requiring a low-potassium diet, such as individuals with chronic kidney disease. The authors conducted a thorough analysis using data from the CFIA, applying appropriate statistical tools, and delivering practical conclusions.
The novelty of the manuscript lies in its use of non-public CFIA data to evaluate the accuracy of potassium labeling—this appears to be the first such study in the Canadian context. An additional strength is the inclusion of a discussion on the potential interactions between sodium-related claims and potassium content. The potassium underestimations identified by the authors could have real clinical implications, thereby justifying the need for further research and potential regulatory review.
At the same time, a few limitations should be acknowledged: the dataset is unbalanced (with over 50% of products being plant-based dairy alternatives), some of the data are over a decade old, and the lack of access to product ingredient lists prevents a deeper analysis of potassium sources (e.g., additives). Therefore, the conclusions regarding the impact of sodium substitutes on potassium content should be interpreted with greater caution.
Response: Thank you for your thorough contentious summary of our paper. We agree that given the data set we need to be cautious with our conclusions. We have updated this in our abstract and conclusion (Lines 33-34 and 36, 474-476).
I kindly ask the authors to consider and address the following suggestions in the revision of the manuscript:
- Please discuss in more detail the overrepresentation of plant-based dairy alternatives in the sample—can this be corrected in the analysis, or at least acknowledged as a significant limitation?
Response: We have expanded further on this in our discussion section (Lines 325-345).
- Please clearly differentiate the results derived from older and newer data (e.g., pre-2010) to improve the relevance and credibility of the conclusions.
Response: This is a great comment and we are so happy we did this analysis as a result of yours and another reviewers comments. The results were very interesting! Thanks for pointing it out. We decided to look at our data pre and post 2017, as the new regulations around potassium labeling came out in December of 2016 (which has now been added to the introduction – Lines 56-60). We explained how we did this analysis in the stats section, added it to the results and discussion section (Lines 151-153, 368-384, 337-345).
- While the references are up to date, you might consider adding studies that assess label accuracy for potassium in other jurisdictions (e.g., the U.S., EU) to strengthen the comparative context. It would be valuable to refer to similar research conducted in other countries (e.g., the U.S., EU) to place the findings in a broader context.’
Response: Our literature review was unable to find many studies that have looked into the accuracy of potassium specifically, however there were two similar papers and as such we have added a section on this to our discussion (Lines 360-368).
- Consider expanding the discussion on the practical implications of potassium underestimation—particularly its potential impact on patients treated with RAAS inhibitors.
Response: This is a great point and valuable add to the discussion. The potential associations of under-reporting potassium content has been more fully added to the discussion (Lines 346-397).
Reviewer 4 Report
Comments and Suggestions for Authors
This manuscript addresses a relevant and underexplored topic with clear clinical implications for patients requiring potassium restriction. However, several important methodological and interpretative concerns must be addressed before the manuscript can be considered for publication.
The potassium measurements from CFIA laboratory analysis form the cornerstone of the study. The manuscript should clearly describe the laboratory methodology, analytical techniqu, quality assurance procedures, and limits of detection/quantification. Without this, reproducibility and interpretation of the results are compromised.
It is unclear whether the distribution of differences between labeled and analyzed potassium values was assessed for normality before applying parametric tests such as paired t-tests and Bland–Altman analysis. This should be explicitly reported, as violation of assumptions could bias results.
Similarly, the chosen cut-off for defining over- or under-reporting (±5 mg) should be justified with a methodological or regulatory reference.
The cut-off or acceptable range for Bland–Altman plots should be stated, with justification.
Performing Bland–Altman analysis for the dataset as a whole may mask product-type-specific biases. Given the heterogeneous nature of the products, separate Bland–Altman plots for major product categories (e.g., plant-based dairy, soups, grains) would be more informative.
The initial dataset contained 17,665 entries, yet only 376 products were included in the final analysis. The authors should provide a detailed discussion on possible causes of missing label potassium values, non-compliance with serving size data.
Moreover, the potential for selection bias is high, as the CFIA sampling strategy is not random and may target certain products more frequently.
The type of product could meaningfully influence discrepancies between labeled and actual potassium content, given variation in natural potassium content and use of potassium-based additives. A subgroup analysis by product type (or at least discussion of observed patterns) would improve interpretability.
While the authors state that ingredient lists were unavailable, it is notable that most products were ultra-processed; this raises the possibility that potassium additives were used. Even without full ingredient lists, brand and product type could be used to infer likely additive use in some cases.
The discussion is very short and refers to a very limited number of references.
It could benefit from exploring potential causes of under-reporting beyond legislation, such as manufacturing variability, potassium loss during storage, or outdated nutrient databases.
International comparisons would strengthen the manuscript, particularly as similar potassium labeling requirements are being introduced in other countries.
Figures should include key statistical annotations (e.g., mean bias and limits of agreement in Bland–Altman plot captions).
Given the potential for high variability in low-potassium products, presenting both absolute and percentage differences side-by-side in tables would provide better context.
Author Response
This manuscript addresses a relevant and underexplored topic with clear clinical implications for patients requiring potassium restriction. However, several important methodological and interpretative concerns must be addressed before the manuscript can be considered for publication.
The potassium measurements from CFIA laboratory analysis form the cornerstone of the study. The manuscript should clearly describe the laboratory methodology, analytical technique, quality assurance procedures, and limits of detection/quantification. Without this, reproducibility and interpretation of the results are compromised.
Response: The methods section explaining the CFIA Compliance Testing has been expanded substantially (Lines 100-124).
It is unclear whether the distribution of differences between labeled and analyzed potassium values was assessed for normality before applying parametric tests such as paired t-tests and Bland–Altman analysis. This should be explicitly reported, as violation of assumptions could bias results.
Response: The decision to use t-tests was decided as evidence has reported that t-tests are the ideal choice when sample sizes are above 200, even for heavily skewed data (Central Limit Theorem). Non-parametric tests have lower statistical power and reduce the granularity (ie, causes loss of information). The outliers in this data set however, are likely clinically meaningful and important reflections of the data.
Reference: Fagerland MW. t-tests, non-parametric tests, and large studies—a paradox of statistical practice?. BMC medical research methodology. 2012 Jun 14;12(1):78.
Similarly, the chosen cut-off for defining over- or under-reporting (±5 mg) should be justified with a methodological or regulatory reference.
Response: This is a great point, we decided to review this threshold and to change it based on the CFIA Compliance absolute amount thresholds, which do allow for a greater variance as the product potassium level increases. We have re-run this data and updated the corresponding Section (section 3.5) and Figure 4 (Lines 154-160, 211-217, Figure 4).
The cut-off or acceptable range for Bland–Altman plots should be stated, with justification. Performing Bland–Altman analysis for the dataset as a whole may mask product-type-specific biases. Given the heterogeneous nature of the products, separate Bland–Altman plots for major product categories (e.g., plant-based dairy, soups, grains) would be more informative.
Response: We have decided to remove the Bland Altman analysis. The distribution of the variances is not normally distributed, when it was broken down into the 9 product subtypes, it became so granular it was hard to get a sense of the data – especially with the uneven distribution of product types. Instead, we are proposing adding a supplemental table, that provides a high level of detail should readers be interested in knowing more.
The initial dataset contained 17,665 entries, yet only 376 products were included in the final analysis. The authors should provide a detailed discussion on possible causes of missing label potassium values, non-compliance with serving size data.
Response: We have added more details regarding this to our results sections to make it clearer (Lines 172, 174-176).
Moreover, the potential for selection bias is high, as the CFIA sampling strategy is not random and may target certain products more frequently.
Response: You are right this is an important limitation in our results. We have added a more robust discussion of this on Lines 325-336.
The type of product could meaningfully influence discrepancies between labeled and actual potassium content, given variation in natural potassium content and use of potassium-based additives. A subgroup analysis by product type (or at least discussion of observed patterns) would improve interpretability.
Response: Yes, we agree with this completely. We have added a new supplemental table that will better enable readers to compare these values by product subtype.
While the authors state that ingredient lists were unavailable, it is notable that most products were ultra-processed; this raises the possibility that potassium additives were used. Even without full ingredient lists, brand and product type could be used to infer likely additive use in some cases.
Response: We have added more about this possibility in the discussion, especially in view of changes in legislation and sodium reformulation strategies that have been published by the Government of Canada between 2005 to 2023 (our data set period) (444-450).
The discussion is very short and refers to a very limited number of references.
Response: The discussion has been expanded on considerably to include more information about legislation, other studies and clinical implications.
It could benefit from exploring potential causes of under-reporting beyond legislation, such as manufacturing variability, potassium loss during storage, or outdated nutrient databases.
Response: When we added a time-based analysis (as was suggested by two other reviewers), we observed a marked difference in the accuracy of labels. We have added this to our discussion as this likely contributes heavily our results (Lines 337-345).
International comparisons would strengthen the manuscript, particularly as similar potassium labeling requirements are being introduced in other countries.
Response: We have added in an study from Australia and commented further on the international context (Lines 360-363)
Figures should include key statistical annotations (e.g., mean bias and limits of agreement in Bland–Altman plot captions).
Response: Bland Altman plot removed. All other figures are presented count-type data. However we added a footnote to Table 2 to better described the statistical testing that was done to prepare this table.
Given the potential for high variability in low-potassium products, presenting both absolute and percentage differences side-by-side in tables would provide better context.
Response: This has been added in the supplemental table.
Round 2
Reviewer 1 Report
Comments and Suggestions for Authors
Make careful revisions according to the reviewer's comments.
Reviewer 2 Report
Comments and Suggestions for Authors
The manuscrip was revised appropriately.
Reviewer 3 Report
Comments and Suggestions for Authors
Dear Authors,
Thank you very much for the clarifications and all significant improvements you made. I find your responses and actions satisfactory.
Best regards,
The reviewer.